# Recent Advances in Oral Vaccines for Animals

**DOI:** 10.3390/vetsci11080353

**Published:** 2024-08-05

**Authors:** Kaining Zhong, Xinting Chen, Junhao Zhang, Xiaoyu Jiang, Junhui Zhang, Minyi Huang, Shuilian Bi, Chunmei Ju, Yongwen Luo

**Affiliations:** 1College of Veterinary Medicine, South China Agricultural University, Guangzhou 510640, China; zhongkaining@stu.scau.edu.cn (K.Z.); xt21823@stu.scau.edu.cn (X.C.); zhangjh@stu.scau.edu.cn (J.Z.); 19803682569@163.com (X.J.); gzzhangjunhuii@163.com (J.Z.); hhminyi@outlook.com (M.H.); 2School of Food Science, Guangdong Pharmaceutical University, Zhongshan 528458, China; shuilianbi@foxmail.com; 3Key Laboratory of Animal Vaccine Development of the Ministry of Agriculture and Rural Affairs, South China Agricultural University, Guangzhou 510640, China

**Keywords:** oral vaccines, mucosal immunity, animal health, infectious disease

## Abstract

**Simple Summary:**

Oral vaccines provide a gentle and convenient way to prevent infectious diseases in animals while inducing robust mucosal immunity that is particularly effective against orally transmitted diseases compared to injected vaccines. However, the harsh environment of the animal’s gastrointestinal tract poses challenges to the delivery of protein antigens to the small intestine, and certain components within the gastrointestinal tract can affect the efficacy of oral vaccines. This review outlines strategies for delivering oral vaccines to the small intestine in animals, examines factors that influence their efficacy, and explores the application of oral vaccines to overcome these challenges and develop optimal solutions for animal vaccination.

**Abstract:**

Compared to traditional injected vaccines, oral vaccines offer significant advantages for the immunization of livestock and wildlife due to their ease of use, high compliance, improved safety, and potential to stimulate mucosal immune responses and induce systemic immunity against pathogens. This review provides an overview of the delivery methods for oral vaccines, and the factors that influence their immunogenicity. We also highlight the global progress and achievements in the development and use of oral vaccines for animals, shedding light on potential future applications in this field.

## 1. Introduction

Vaccines play a critical role in the prevention of infectious diseases in humans and animals. The use of vaccines has effectively controlled animal diseases caused by pathogens such as foot-and-mouth disease virus (FMDV), classical swine fever virus (CSFV), infectious bursal disease virus (IBDV), and even led to the eradication of rinderpest virus [1,2,3,4]. Traditional injectable vaccines involve the administration of vaccine components into the skin or muscles using a needle, eliciting a robust systemic immune response that confers immunity to infections. However, injectable vaccines have inherent drawbacks. For instance, the use of needles to administer vaccines can cause stress reactions in animals, posing risks to veterinarians. Additionally, the use of contaminated needles carries the potential for spreading bloodborne diseases and incurring losses. Furthermore, zoonotic diseases have become a global public health concern because infections originating in animal populations can be transmitted to humans through fecal contamination, slaughter and direct contact, leading to the spread of infectious diseases in human populations. To prevent and control infectious diseases at their source, vaccination of susceptible animal populations with appropriate vaccines is essential. Nevertheless, controlling infectious diseases in wildlife, a particularly susceptible animal group, presents challenges. Trapping and injecting vaccines into wild animals can disrupt their normal activities and entail significant manpower and costs, making it an impractical approach.

Approximately 90% of pathogens invade body tissues through mucosal routes, including the gastrointestinal, respiratory, and genitourinary systems [5]. These mucosal surfaces are particularly vulnerable due to their larger exposed area and thinner, more permeable barrier compared to the skin. Therefore, robust mucosal immune responses are crucial as the primary defense against pathogens and their toxins affecting mucosal surfaces [5]. The oral vaccine is a special type of vaccine that induces mucosal immune response by introducing the active ingredients into the body through oral administration. Due to their ease of use, high compliance, and superior safety profile, oral vaccines offer significant advantages in bolstering immunity in both livestock and wildlife. The delivery vector of oral vaccines serves to protect the antigen from the low-pH environment and proteases of the digestive tract and effectively transports the active components of the vaccine to the small intestine. In the small intestine, these active components are exposed, recognized, and subsequently ingested by microfold cells (M cells) at Peyer’s patches (PPs) on the intestinal wall (Figure 1). Antigen-presenting cells (APCs), such as dendritic cells (DCs), present antigens to CD4^+^ T cells in the gut-associated lymphoid tissue (GALT), thereby inducing mucosal humoral immunity and cellular immune responses. In addition, monocytes located on the basal side of the intestinal mucosa can directly take up antigens from the intestinal cavity, leading to the induction of a mucosal immune response. The antibodies produced by mucosal humoral immunity following oral vaccination are mainly secretory IgA (sIgA) and IgG. sIgA is essential for the development of the mucosal barrier, being predominantly located in the outer layer of the mucosa. Its primary functions include binding and clustering of pathogens, blocking their entry into mucosal epithelial cells, restricting their movement, and eliminating pathogens present in the mucosa [6,7,8]. In particular, secretory IgA antibodies are critical for resistance to gastrointestinal pathogens, with studies demonstrating that mice can withstand ten times the infectious dose of Shiga toxin-producing *Escherichia coli* following a single dose of oral *E. coli* vaccine that induces the production of secretory IgA but not serum antibodies [9].

Oral vaccines contain a variety of types, including oral attenuated vaccines, recombinant vectored oral vaccines, nanoparticle oral vaccine (subunit or nucleic acid vaccines), and transgenic plant oral vaccine [10] (Table 1). The introduction of the first oral vaccine against poliovirus in the 1960s marked a significant milestone in the fight against infectious diseases [11]. This pioneering development paved the way for the subsequent creation of oral vaccines targeting various gastrointestinal pathogens, including rotavirus, *Vibrio cholerae*, and typhoid [10,12]. Despite these advances, the oral delivery of vaccines encounters numerous obstacles within the digestive tract, such as degradation and malabsorption issues caused by mucosal barriers. Consequently, the evolution of oral viral vaccines has been gradual, with the range of currently authorized oral vaccines for humans confined to those that address diseases impacting the gastrointestinal tract. These vaccines are primarily of the live attenuated or inactivated varieties. Furthermore, alternative vaccine formulations, such as oral subunit and nucleic acid vaccines, face significant challenges due to their sensitivity to degradation. These challenges make them more difficult to administer orally, and as a result, they have not yet reached the commercialization stage. Currently, the majority of animal oral vaccine research is focused on the prevention of rabies and tuberculosis in wildlife and urban strays, with additional efforts directed toward the development of oral vaccines for the prevention of coccidia, parvovirus, pseudorabies virus, and other diseases. In this review, we provide a comprehensive discussion of the types of delivery vectors used in oral vaccines for animals, the factors influencing the efficacy of oral vaccines, and the research progress made in various oral vaccine developments. In addition, we present the potential applications of oral animal vaccines around the world.

## 2. Considerations for Designing Oral Vaccines for Animals

### 2.1. Challenges of the Harsh Gastrointestinal Environment

The animal digestive system consists of several key components, including the oral cavity and its associated structures such as the lips, teeth, tongue, and salivary glands; the esophagus; the specialized stomachs (reticulum, rumen, and omasum) found in ruminants and the true stomach in all species; followed by the small intestine. In addition, there are essential organs such as the liver, the exocrine portion of the pancreas, the colon, and the final parts of the system—the rectum and anus. In addition, the presence of GALT, which includes structures such as tonsils, Peyer’s patches, and diffuse lymphoid tissue, highlights the immune functionality along the gastrointestinal tract [19]. The structures and immune functions of the digestive systems vary greatly among different animals, such as omnivores, herbivores, carnivores, and avian species. Therefore, when designing or selecting oral vaccines, it is essential to tailor them specifically to the characteristics of each animal species. Following ingestion by animals, oral vaccines must withstand various environmental challenges in the digestive system, including exposure to the highly acidic conditions of the stomach and degradation by enzymes, before reaching the small intestine where they can induce a mucosal immune response. Vaccines using gut microbes as carriers can survive in the extreme environment of the gastrointestinal tract and stimulate mucosal immune responses [10]. For easily degradable protein subunit vaccines and nucleic acid vaccines, it is necessary to incorporate appropriate nano-delivery carriers to improve their stability and targeting in the gastrointestinal tract. In addition, oral immunization requires a higher dose of antigen to induce an immune response compared to parenteral immunization [20], but higher doses of immunogens also increase the likelihood of inducing immune tolerance [21]. Therefore, oral vaccine delivery vehicles typically require adjuvant functionality to adequately stimulate the immune system [22].

### 2.2. Microfold Cell

Microfold cells, situated on the surface of the small intestinal mucosa, exhibit a distinct morphology compared to the small intestinal villous epithelial cells. With an abundance of microfolds, M cells primarily function in the uptake and transport of exogenous antigens. These cells feature a U-shaped “pocket” structure at the basal level of the intestinal mucosa, housing a substantial population of lymphocytes, including T lymphocytes, B lymphocytes, dendritic cells, and macrophages. This unique configuration enables M cells to efficiently deliver antigens to the germinal centers of intestinal-associated lymphoid tissues, thereby eliciting specific immune responses [23].

Leveraging the presentation of intestinal antigens by intestinal immune cells through M cells, oral vaccines can be strategically formulated with delivery vectors or adjuvants tailored to target M cells. Notably, Tetragalloyl-D-lysine dendrimer (TGDK) emerges as a promising M cell-targeting molecule, capable of traversing the gastrointestinal tract and translocating to the basolateral side of the intestines via M-like cells. Subsequently, TGDK accumulates in the germinal centers, thereby substantiating its potential utility as a targeting molecule for M cells [24].

The researchers have identified the short peptides CKS9 (amino acid sequence: CKSTHPLAC) and Co1 (amino acid sequence: SFHQLPARSPLP) as M cell-targeting molecules using the phage display screening technique. These peptides are coupled with water-soluble chitosan, facilitating the targeted delivery of oral vaccines to Peyer’s patches. Furthermore, CKS9 exhibits potential for co-expression with IL-6 on the surface of *Lactobacillus* IL1403, presenting a novel mucosal adjuvant capable of eliciting both Th1 and Th2 type immune responses. This co-expression results in the production of specific IgA and IgG antibodies, along with Th2-type immune response cytokines IL-6 and IFN-γ [25,26]. The Co1 short peptide was employed to bind foot-and-mouth disease (FMD) viral proteins and porcine epidemic diarrhea virus (PEDV) through a linker peptide to generate a fusion protein, subsequently displayed on the surface of *Lactobacillus* spp. Notably, the outcomes of these experiments revealed that the inclusion of the Co1 peptide led to the enrichment of oral vaccines in Peyer’s patches, accompanied by heightened specific antibody titers and increased concentrations of multiple serum cytokines [27].

### 2.3. Gut Microorganisms

The gut bacterial flora plays a pivotal role in regulating the intestinal tract and various physiological functions of the organism. Recent research across multiple disciplines has shed light on the interactions between the gut microbiota and other systemic functions, such as the brain–gut axis [28]. Given that the small intestine serves as the primary site of action for oral vaccines, the presence of these vaccines may potentially impact the balance of the gut microbiota, leading to dysbiosis and disruption of normal physiological functions, thereby posing a potential safety concern. Furthermore, the status of the gut microbiota may also influence the immunization efficacy of oral vaccines. Consequently, several scholars have initiated investigations into the relationship between oral vaccines and the gut microbiota.

In the field of research investigating the impact of gut flora on the immune response to oral vaccines, a review conducted by Petra Zimmermann [29] highlighted the lack of studies specifically investigating the relationship between human gut flora and oral vaccines before 2018. The findings from these studies indicated that the relative abundance of the Firmicutes and Actinobacteria correlated with higher humoral and cellular immune responses, while the relative abundance of the phylum Proteobacteria and Bacteroidetes was associated with lower immune responses.

To date, the exploration of the relationship between oral vaccines and gut microbiota remains relatively underexplored, representing an open area of investigation. The molecular mechanisms through which oral vaccines influence gut microbiota and vice versa have yet to be fully elucidated. A comprehensive understanding of these mechanisms will establish a theoretical foundation for safeguarding the overall health of both animal and human organisms, while maximizing the efficacy of oral vaccines.

### 2.4. Adjuvants

Oral vaccine adjuvants encompass substances capable of augmenting the immunogenicity of antigens upon entering the body concurrently or prior to the oral vaccine. Traditional adjuvants, such as propolis, aluminum gum, and oil-based adjuvants, have been extensively utilized in injectable vaccines for animals, leveraging their facile preparation and well-established technology. However, the applicability of aluminum gum and oil adjuvants in oral vaccines is constrained by the disparity between the oral and injectable administration routes. Conversely, propolis adjuvants have garnered increased attention in oral vaccines, yielding favorable outcomes [30]. 

In addition to traditional adjuvants, novel adjuvant variants have emerged in recent years, diverging from traditional adjuvants in terms of source, immunogenicity enhancing mechanisms, and preparation methods. These include microbial components, cytokines, and nano-adjuvants, among others. Notably, microbial components and cytokines can not only be combined with antigens for oral immunization but can also be genetically engineered by incorporating genes encoding microbial components or cytokines into the vectors of recombinant bacteria and viruses. Subsequently, these genes can be expressed to realize their adjuvant effects, thereby broadening the spectrum of adjuvant options for oral vaccines.

#### 2.4.1. Microbial Components Adjuvant

Microbial component adjuvants, as immunogenic constituents of microorganisms, particularly bacteria, are capable of eliciting an immune response following bacterial infection of the host organism. For instance, oral immunization with *Escherichia coli* heat-labile enterotoxin B subunit (LTB) significantly enhances the humoral immunity of the host [31]. 

Bacterial flagellin, a pathogen-associated molecular pattern (PAMP), represents a highly conserved molecular pattern capable of eliciting a natural immune response in animals. This molecule is recognized by pattern recognition receptors (PRRs), which subsequently trigger a series of downstream immune responses. Notably, Toll-like receptor (TLR) serves as a prominent type of PRR, with bacterial flagellin acting as an agonist of TLR5. Upon activation, TLR5 induces the maturation of naïve human dendritic cells and the expression of diverse immune molecules [32]. Research findings indicate that the incorporation of the bacterial flagellin gene into the LBNSE strain of rabies virus using genetic engineering methodologies can lead to a 90% protection rate in mice, as opposed to the 50% protection rate observed with the LBNSE strain alone. This compelling evidence underscores the substantial enhancement of immunogenicity achieved through the introduction of the bacterial flagellin gene as an adjuvant in the LBNSE strain of rabies virus [33].

In addition to TLR5, agonists of TLR9 present a viable option as microbial component adjuvant. TLR9 is capable of recognizing unmethylated cytosine guanine dinucleotides (CpG sequences) present in viruses and bacteria, thereby eliciting an adjuvant effect that enhances the immune response through the activation of downstream B cells, T cells, NK cells, and APCs [34,35]. Building upon this premise, researchers have leveraged synthetic CpG oligodeoxynucleotides (CpG ODN) to selectively stimulate TLR9 in immune cells, thereby enhancing adaptive immune responses as well as innate responses systemically and in the intestine [36]. This approach holds promise as an adjuvant for oral subunit vaccines.

Outer membrane vesicles (OMVs) derived from Gram-negative bacteria primarily serve as key components in the synthesis of bacterial outer membranes, vesicular transport, nutrient acquisition, and intercellular communication. Their rich content of PAMPs renders them particularly suitable for utilization as adjuvants in oral vaccines. The substantial impact of OMVs as adjuvants for oral vaccines is underscored by the remarkable effectiveness observed upon their incorporation into inactivated oral vaccines protect against *Salmonella*, with the addition of OMVs from *Burkholderia pseudomallei* resulting in a noteworthy increase in the protection rate from less than 25% to 75% [37].

#### 2.4.2. Cytokine Adjuvant

Within the cytokine family, interleukins (ILs) represent a diverse class of cytokines secreted by leukocytes and other immune cells, playing pivotal roles in mediating interactions among various immune cells. Comprising a wide array of styles, interleukins, including IL1–IL35, are associated with a broad spectrum of functions. Notably, IL-2 is recognized for its capacity to promote the proliferation and differentiation of B cells, T cells, and NK cells, rendering it a valuable adjuvant for oral DNA vaccine protect against foot-and-mouth disease and other subunit vaccines [38,39,40].

Colony-stimulating factors (CSFs) constitute a class of cytokines that exhibit the selective ability to stimulate hematopoietic stem cells, thereby guiding their differentiation into specific immune cells. Among these, granulocyte monocyte colony-stimulating factor (GM-CSF) is widely employed as an adjuvant in oral vaccines, owing to its capability to induce the differentiation of hematopoietic stem cells into monocytes and enhance the ability of uptake and process the antigens by immune system. Notably, the introduction of the GM-CSF gene into the LBNSE strain of rabies virus led to the development of the recombinant virus LBNSE-GMCSF, resulting in an oral immunization protection rate exceeding 80% in mice. This rate not only surpassed that of the parental LBNSE attenuated strain but also rivaled that of the flagellin adjuvant [33].

The effects of cytokines often demonstrate synergistic, superimposable, and species-specific characteristics. Consequently, when employing a cytokine as an adjuvant in oral vaccines, it is imperative to consider the synergistic effects of the cytokine and select a cytokine from the target animal species. Research has revealed that the combined use of IL-2 and GM-CSF yields superior results compared to their individual use. Moreover, the co-expression of IL-2 and GM-CSF exhibits a more pronounced vaccine effect than the combined vaccination of IL-2 and GM-CSF, thus demonstrating the favorable adjuvant effect of their synergistic use [39]. Future investigations could delve into exploring additional cytokine combination adjuvants capable of eliciting robust humoral and cellular immunity concurrently for application in oral vaccines.

### 2.5. Bait

For the successful immunization of wild or stray animals using oral vaccines, the use of baits is essential to attract wildlife to consume the oral vaccines. These oral vaccine baits must possess specific attributes, including ease of ingestion by wildlife, non-interference with the immunogenicity of the oral vaccine, and resilience to adverse field conditions. Researchers have proposed various options for bait design to address these requirements.

#### 2.5.1. Conventional Bait

Historically, conventional baits have been tailored to align with the dietary preferences of wildlife, encompassing formulations such as meat powder, milk powder, intestine baits, and fish bait. However, the diverse selection of baits employed across various studies conducted worldwide renders it challenging to ascertain the most effective traditional baits for oral vaccine delivery to animals. Recent investigations, nonetheless, have highlighted fish bait and intestine baits as notably more effective options for oral vaccine baiting [41].

#### 2.5.2. Novel Bait

The innovative bait is distinguished by its capacity to elicit consumption by a particular species (species-specific). Notably, two commercial bait companies in New Zealand have developed a paste bait that appeals to European badgers. This paste bait has been combined with an edible vaccine carrier derived from hydrogenated peanut oil to formulate an oral vaccine targeting *Mycobacterium bovis*. Furthermore, the bait has demonstrated environmental resilience, retaining its efficacy for up to 7 days at temperatures below 20 °C. However, the specific composition of the paste bait has not been publicly disclosed [42]. In the event that an oral vaccine tailored for raccoons and skunks is to be developed, the use of fish oil baits could be considered [43]. Despite the potential of these two novel baits to entice consumption by specific animals, their actual effectiveness in facilitating oral immunization within the targeted area remains to be evaluated.

#### 2.5.3. Biomarkers of Bait

In the realm of research methodology, the majority of studies investigating bait ingestion rates have relied on the use of biomarkers. One of the most prevalent biomarkers is the bloodborne iophenoxic acid (IPA), initially developed as a cholecystography agent in the previous century owing to its minimal side effects and low systemic toxicity. Its protracted residence time in the body and ability to enter the animal’s bloodstream render it instrumental in determining whether an animal has consumed oral vaccine bait [44]. However, a drawback of this bait marker is its reliance on blood collection for detection, a procedure that poses stress to wild animals, thereby complicating research efforts and conflicting with animal welfare principles. In response, researchers have explored two novel animal-friendly baiting markers that obviate the need for detection methods that may induce stress in animals. One marker is detectable through canine feces [45], while the other entails observing the color of the dog’s oral mucosa, specifically, the lingual mucosa [46]. Notably, the latter can be remotely observed by inspecting the color of a dog’s tongue, eliminating the need for direct contact with or trapping of the animal. Nevertheless, the potential impact of these specialized dyes and plastic balls on the socialization and intestinal health of the dog warrants comprehensive exploration.

#### 2.5.4. Bait Deployment Strategies

Apart from bait formulation, various other factors exert influence on the consumption of oral vaccine baits by wildlife. Research findings have indicated that the uptake and absorption rates of oral vaccine baits are higher during the spring compared to the summer. Additionally, oral vaccine baits exhibit lower uptake and absorption rates within social groups of animals, potentially attributable to intra-group competition for baits. Moreover, the uptake and absorption rates of oral vaccine baits appear to be unaffected by the age of the animal and whether the bait is positioned on the ground. Consequently, it is recommended that oral vaccines be deployed in the wild during the spring or early summer to optimize uptake and absorption rates [47].

## 3. Delivery Strategies and Vectors of Oral Vaccines

### 3.1. Classical Delivery Strategies

Traditional oral vaccines for animals are primarily derived from live attenuated strains of pathogens or use viral/bacterial carriers. They are typically administered through feed or drinking water, such as *Lawsonia intracellularis*, *Erysipelothrix rhusiopathiae*, and rotavirus vaccines in pigs and protection against Newcastle disease, infectious laryngotracheitis, and avian encephalomyelitis in poultry [48]. 

Most oral vaccines primarily trigger mucosal immune responses in the GALT located in the small intestine, particularly in Peyer‘s patches. It is worth noting that certain attenuated oral vaccines can also elicit effective immune responses in the tonsils. For instance, a study of an oral attenuated rabies vaccine demonstrated the absence of virus-infected cells in the palatine tonsils of skunks, suggesting that less efficient uptake or infection of the vaccine virus by skunks may have resulted in a diminished response to the oral vaccine [49]. Another study demonstrated that the attenuated rabies vaccine strain primarily replicated in the tonsils of dogs following oral administration, indicating that the tonsils may play a pivotal role in the uptake of the attenuated rabies vaccine strain [50]. Understanding the mechanism of uptake of different attenuated oral vaccines in tonsils will help in the future development of novel oral vaccines and adjuvants.

However, not all types of vaccines are suitable for oral immunization. In particular, subunit vaccines and nucleic acid vaccines without a carrier cannot be effectively administered by the oral route. Therefore, alternative delivery methods are needed to ensure their efficacy and protective effects.

### 3.2. Vectors of Nanoparticles

Nanoparticles play a pivotal role in oral vaccines by encapsulating protein antigens and shielding them from the harsh conditions of the gastrointestinal tract. Once administered orally, these nanoparticles can protect the antigens during their passage through the digestive system.

Upon reaching the intestine, nanoparticles facilitate the delivery of antigens to the GALT. They achieve this by interacting with M cells and other enterocytes, which help transport the antigens across the intestinal lining. Once inside the GALT, the antigens are released and taken up by antigen-presenting cells (APCs). This process is crucial for initiating a targeted immune response [51]. 

#### 3.2.1. Poly Lactic Acid (PLA) and Poly Lactide-Co-Glycolide Acid (PLGA)

PLA and PLGA are biocompatible synthetic materials that can be taken up by APCs in vivo. Meanwhile, PLA and PLGA are positively charged, which can bind to negatively charged DNA and target it for delivery to the APCs and induces specific immune responses in the body. Particularly, PLGA is an enteric delivery system approved by the FDA and European Medicines Agency with carrier and adjuvant properties that protect antigen from hydrolysis, target M cells, and slow release of antigen, which has been used to deliver oral vaccine antigens in more studies [52].

#### 3.2.2. Chitosan Nanoparticle

Chitosan is a natural polymer that is adhesive, biodegradable, and non-toxic. More importantly, nanoparticles made of chitosan can be stabilized under acidic conditions, releasing 14% of proteins at pH = 5.5, and it cumulatively releases 75% of proteins at pH = 7.7 [53], which ensures that the active ingredients of the vaccine can be protected from gastric acid and protease.

#### 3.2.3. Liposome 

The structure of the liposome is similar to the cell membrane. Depending on the hydrophilic and hydrophobic nature of the antigen, the location of the antigen binding to the liposome is different. The hydrophilic antigen is bound to the outside of the liposome, while the lipid soluble antigen is embedded in the inside of the lipid bilayer. Liposome can target specific cell by adding the ligand to the surface of liposome, and can control the release of antigens in animals may continuously induce immune responses [54,55].

#### 3.2.4. Bilosome

Bilosome has a similar structure to liposomes, with bile salts in the middle of the lipid bilayer of bilosomes to enhance their stability; thus, bilosomes are more stable and resistant against the harsh environment of the gastrointestinal tract compared to liposomes. Bilosomes usually use glucomannan-modified bilosomes (GM-bilosomes), which is often used as ligands for targeting M cells and also resist the degradation of antigens by digestive hydrolytic enzymes [56,57]. Related studies also show that GM-bilosomes can induce higher IgG titers in the body than those induced by bilosomes [58]. 

#### 3.2.5. Virus-like Particles (VLPs)

VLPs are created using viral structural proteins that can self-assemble into particle structures but lack viral genomic material, rendering them non-infectious. VLPs can be produced using various expression systems, including bacteria, yeast, insect cells, and mammalian cells. While mammalian cells are ideal for generating high-quality VLPs with structures similar to those found in nature, the use of insect cells is often preferred in practice due to cost considerations and the ability to produce effective antigens [59]. For instance, an avian influenza oral vaccine utilizing haemagglutinin (HA) VLPs produced in silkworms demonstrated the ability to induce hemagglutination inhibition (HI) antibodies in both mice and chickens following oral administration [59]. The variant-specific surface protein (VSP) on *Giardia lamblia* trophozoites exhibits remarkable resistance to proteolytic digestion, pH extremes, and high temperatures [60]. This protein plays an important role in stimulating the host innate immune response through a TLR-4-dependent pathway. Chimeric virus-like particles (VLPs) decorated with VSP and HA exhibit enhanced stability and are effective in activating antigen-presenting cells in vitro. When administered orally, these VSP-pseudotyped VLPs elicit potent immune responses that provide protection against influenza infection in mice [60].

### 3.3. Recombinant Bacterial and Yeast Vectors

With the advancement and refinement of genetic engineering technology, the manipulation of bacterial genomes has become more feasible. Among various bacterial strains, Lactobacillus stands out as an ideal carrier for the delivery of oral vaccines. Lactobacillus is a probiotic bacterium that naturally colonizes the intestinal tract. Several studies have investigated the use of Lactobacillus as an carrier of oral vaccines for animals, demonstrating its efficacy in inducing mucosal immune responses against rabies virus (RABV) and PEDV [61,62,63]. These vaccines have induced specific neutralizing antibodies, cytokine production, and lymphocyte proliferation, suggesting their potential for practical application and industrialization.

*Salmonella* is a versatile recombinant bacterial vector that has been used extensively in various research studies. These attenuated strains are capable of carrying exogenous antigens or DNA and infecting APCs (antigen presenting cells) in the intestinal diffuse lymphoid tissues via infected M cells [64]. Deletion of the *asd* gene in *Salmonella typhimurium* results in the loss of the ability to synthesize diaminopimelic acid (DAP), a unique and essential component of the rigid layer of the bacterial cell wall. The absence of DAP leads to defective replication of *Salmonella*, ensuring the safety of the *Salmonella* vector in the body without causing disease [65]. The use of *Salmonella* to express antigenic proteins of H9N2 avian influenza virus in an oral vaccine has shown promising results in animal studies. Chickens not only developed immunity to H9N2, but also showed a high level of protection against *Salmonella*. These results suggest that oral vaccines using *Salmonella* vectors have the potential to simultaneously reduce the risk of enteric disease [66].

Yeast, a fungus that is non-pathogenic to animals, can be effectively used in vaccine development due to its resilience in acidic environments such as the stomach. This allows for yeast to deliver oral vaccine antigens and adjuvants to the small intestine, where they can induce both mucosal and systemic immune responses [67,68]. Using the yeast surface display platform, antigenic proteins can be anchored to the surface of yeast cells, which then travel to the small intestine to stimulate immune responses [69]. Alternatively, with the yeast cell expression and secretion platform, antigenic proteins can be secreted from the yeast cells once they reach the small intestine, making it an ideal carrier for oral subunit vaccines [70].

### 3.4. Recombinant Virus Vectors

#### 3.4.1. Recombinant Poxvirus Vector

Poxvirus, the largest known animal virus with a double-stranded DNA genome, offers unique advantages for vaccine development. Unlike other DNA viruses, poxviruses replicate in the cytoplasm of infected cells, ensuring that their DNA remains nontransmissible and relatively safe. A notable example of a recombinant oral vaccine using poxviruses as vectors is the RABORAL V-RG^®^ vaccine. This vaccine effectively incorporates rabies virus glycoprotein into the cowpox virus. Since its introduction in 1987, the V-RG^®^ vaccine has demonstrated an exceptional safety profile, high protective efficacy, and has successfully eliminated rabies from certain regions of the state of Texas in field applications [71]. However, it is worth noting that from 2001 to 2009, 6 cases of adverse reactions due to human exposure to V-RG^®^ vaccine and 59 cases of pet exposure were reported in the U.S. Of the human cases, one person required hospitalization, while none of the pet cases were serious [72]. Although these numbers are relatively low, measures to minimize both human and pet exposure to V-RG^®^ vaccine are necessary to ensure its safe use.

#### 3.4.2. Recombinant Adenovirus Vector

Adenoviruses (AdVs) are a common class of viruses in animals and humans, with a diameter of approximately 70–90 nm. These viruses have a genetic material consisting of double-stranded DNA, lack an envelope, and adopt an icosahedral shape. Due to their advantageous properties, including lack of replication after modification, large capacity for insertion of exogenous genes (up to 36 kb), cost-effective industrial mass production, non-integration into the host genome, and ability to elicit a robust immune response without the need for adjuvants, adenoviruses have been extensively used in experimental applications. In particular, they are widely used as vaccine vectors against infectious diseases, as viral vectors for the treatment of tumors, and as gene therapy vectors. When recombinant adenoviruses are used as vectors, a single dose of immunization can induce strong humoral and cellular immunity [73]. 

Since AdVs are widely present in nature, animals may have been infected with AdVs prior to exposure to recombinant oral adenovirus vaccine, and specific antibodies to AdVs were presented in the body, the immunizing effect of recombinant oral adenovirus vaccine on the animal will be limited, which is an effect known as pre-immune influence. Pre-immunization effects are a major obstacle to the promotion of oral adenovirus-vectored vaccines, and current strategies to counteract pre-immunization effects in human injectable vaccines include the use of heterologous adenovirus vectors, increasing the vaccine dosage, and administering a secondary booster vaccination [14,74,75].

Among oral recombinant adenovirus vaccines for animals, ONRAB^®^ is an oral recombinant rabies virus glycoprotein based on human adenovirus type 5 as a vector (AdRG1.3), and the recombinant virus, which is replication-competent, was developed for rabies immunization of wildlife. The vaccine has been used in Ontario, Canada since August 2006 [76]. Several studies have shown that ONRAB^®^ recombinant virus has a low risk of horizontal transmission, a good safety profile, and good protection against a wide range of wildlife, including raccoons, red foxes, and skunks [73,77,78,79,80], making it a good recombinant adenoviral oral rabies vaccine for use in wildlife. Besides that, canine type II adenovirus has been used as a vector. The oral vaccine CAV-2-E3Δ-RGP strain was prepared by knocking out its E3 gene and importing the G protein-coding gene of rabies virus. Using this strain to orally immunize ferret badgers, 85% of ferret badgers had rabies virus-neutralizing antibody titers in serum higher than 0.5 IU/mL, which was better than the attenuated strain of rabies virus, SRV9, proving the effectiveness of the heterologous adenovirus vector [81]. However, the use of canine type II adenoviral vectors for oral immunization of dogs is susceptible to the effects of pre-immune antibodies, and strategies to resist the effects of oral immunization pre-immunization remain to be explored.

### 3.5. Transgenic Plant

Oral vaccines utilizing transgenic plant vectors involve the synthesis of antigenic proteins within plant cells via transgenic technology. Animals consume these plants directly, leading to the development of oral immunity. The inherent properties of the plant cell wall, which naturally resist gastric acids, protect it from decomposition by microorganisms until it arrives in the small intestine [16]. There, the release of antigens occurs, effectively triggering immune responses. This approach aligns with the principles of oral subunit vaccines. The application of oral vaccines prepared with transgenic plant vectors holds great promise for swine farms. Given that a majority of swine feed is plant-based, these vaccines can be effectively employed to combat intestinal infectious diseases in swine, including *E. coli* and PEDV [82,83].

## 4. Utilizations of Oral Vaccines in Various Animals

### 4.1. Oral Vaccines for Urban Stray Animals and Wildlife

Urban stray animals and wildlife, including dogs, ferrets, badgers, foxes, and skunks, often serve as reservoirs for pathogens such as the rabies virus and *Mycobacterium bovis*. These pathogens pose a significant risk of transmission to humans and can lead to severe diseases. Implementing oral vaccines for these animals offers a practical approach to reducing the spread of such infections. By targeting these at-risk populations, we can mitigate the potential public health threats posed by these pathogens. We summarized the types and protective rate of candidate of oral vaccines for urban stray animals and wildlife, respectively, in Table 2.

Oral attenuated vaccines, characterized by transient pathogen replication within the host organism, typically exhibit robust immunogenicity without necessitating adjuvant supplementation, and they are widely used for urban stray animals and wildlife. Nonetheless, oral attenuated vaccines pose inherent risks, including horizontal or vertical transmission in natural settings and the potential for virulence reversion. However, research indicates that the SPBN-GASGAS strain of the rabies virus presents minimal risks of horizontal and vertical transmission in various animal species, including foxes, raccoon dogs, mongooses, striped skunk, domestic dogs, domestic cats, and domestic pigs; therefore, this strain demonstrates favorable environmental safety profiles [84,85]. Further refinement of oral attenuated vaccines remains imperative due to inherent limitations such as the environmental susceptibility of attenuated strains and the necessity for heat-resistant variants. Notably, the most recent iteration of the oral attenuated rabies virus strain, SPBN-GASGAS, exhibits diminished temperature stability above 20 °C [86]. This susceptibility may compromise immunogenicity during delivery, potentially impeding successful immunization outcomes. Efforts to address these challenges are crucial for enhancing the efficacy and reliability of oral attenuated vaccine platforms.

**Table 2 vetsci-11-00353-t002:** Oral vaccine candidates for urban stray animals and wildlife.

Pathogen	Trial Animals	Type	Protection Rate	Strain/Vector	Antigen	References
Rabies virus	Mice	Attenuated	90%	LBNSE-GMCSF	-	[33]
	Mice	Attenuated	90%	LBNSE–flagellin	-	[33]
	Mice	Attenuated	100%	ERAG3G	-	[87]
	Ferret badgers	Attenuated	-	SRV9	-	[81]
	Mice	Attenuated	100%	ERA	-	[88]
	Canine	Attenuated	100%	LBNSE-dGM-CSF	-	[50]
	Wolves	Attenuated	-	SAG2	-	[89]
	Foxes	Attenuated	89.6%	SPBN GASGAS	-	[90]
	Foxes, raccoon dogs	Attenuated	>90%	SPBN GASGAS	-	[91]
	Canine	Attenuated	-	SPBN GASGAS	-	[92]
	Foxes, raccoon dogs	Attenuated	-	SPBN GASGAS	-	[93]
	Mice	Attenuated	90%	LBNSE-U-OMP19	-	[94]
	Red foxes, raccoon dogs	Attenuated	-	SAD–Bern	-	[95]
	Dogs	Attenuated	-	SPBN GASGAS	-	[96]
	Mice	Recombinant vector	50–60%	*Salmonella*	Glycoprotein	[97]
	Red foxes	Recombinant vector	33–62%	Adenovirus	ONRAB^®^	[98]
	Skunks	Recombinant vector	81–100%	Adenovirus	ONRAB^®^	[80]
	Ferret badgers	Recombinant vector	-	Adenovirus	Glycoprotein	[81]
	Mongooses	Recombinant vector	-	Adenovirus	ONRAB^®^	[99]
	Mice	Recombinant vector	60%	*Lactobacillus*	Glycoprotein	[61]
	Goats, foxes	Recombinant vector	-	Newcastle disease virus	Glycoprotein	[100]
*Mycobacterium bovis*	Wild boar	Inactivated	-	-	-	[101]
	Red deer	Inactivated	-	-	-	[102]
	Wild badger	Attenuated	-	Liporale–BCG	-	[103]
Ebola virus	Chimpanzees	Recombinant vector	-	RABV	Glycoprotein	[104]
*Bacillus Anthracis*	Mice	Attenuated	-	34F2 spores	-	[105]
*Toxoplasma gondii*	Mice	Plant-based	-	Tobacco	HSP90-SAG1	[17]
*Chlamydia muridarum*	Mice	Attenuated	-	CMmut/IntrOv	-	[106]

### 4.2. Oral Vaccines for Economic Animals

Economic animals, including pigs, cattle, and poultry, are frequently raised on a large scale, and their products significantly contribute to the human food supply. Consequently, animal husbandry is a substantial industry where animal health directly affects farm profitability. Additionally, since most livestock farming is large-scale, using injectable vaccines demands significant human and material resources. The practice of using shared needles for vaccination in some farms can also the spread of bloodborne pathogens. Therefore, applying oral vaccines in livestock operations can lead to considerable cost savings. We have summarized the types and protective rates of candidate oral vaccines for economic animals in Table 3.

The expansion of large-scale, intensive pig farming operations has underscored the critical importance of effective vaccination in the swine industry. While traditional routes of vaccine administration have involved subcutaneous or intramuscular injections, the rise of extensive pig farming enterprises has catalyzed a transition to alternative mass vaccination techniques. In particular, oral vaccination has gained traction as an increasingly preferred method in swine management practices [107]. Currently, commercialized oral vaccines for swine primarily target rotavirus and *Erysipelothrix rhusiopathiae*. Meanwhile, vaccines still in research focus on PEDV, CSFV, pseudorabies virus (PRV), and pathogenic *Escherichia coli* [108] (Table 3).

Poultry farming is an important industry that is significantly threatened by various pathogens, both zoonotic and non-zoonotic. Drinking water vaccination has become the most commonly used method of immunization in modern poultry production. This method typically involves the use of attenuated vaccines, such as those for Newcastle disease, infectious bursal disease, infectious bronchitis and salmonellosis in chickens, and viral hepatitis vaccines in ducks. In addition, there is an increasing focus on the development of oral vaccines against avian influenza virus (AIV) and poultry parasites such as chicken coccidia and *Eimeria tenella* [109] (Table 3).

Regarding the types of oral vaccines for livestock, oral subunit vaccines and oral nucleic acid vaccines represent innovative paradigms in oral vaccination. While the active constituents of oral subunit vaccines consist of antigenic epitopes from pathogens along with adjuvants, oral nucleic acid vaccines operate on a novel principle. However, due to the exigencies of their preservation, both vaccines face limitations in application scenarios. Moreover, their effective duration upon release into the environment is comparatively shorter than that of oral attenuated vaccines and oral vaccines with vector. Consequently, future research endeavors should prioritize the development of techniques for enhancing the heat resistance of subunit antigens and nucleic acids, thereby ensuring the sustained effectiveness of oral subunit vaccines and oral nucleic acid vaccines under the harsh environmental conditions encountered in natural settings.

**Table 3 vetsci-11-00353-t003:** Oral vaccine candidates for economic animals.

Pathogen	Trial Animals	Type	Protection Rate	Strain/Vector	Antigen	References
PRV	Domestic pigs	Attenuated	100%	Bartha	-	[110]
CSFV	Pigs	Attenuated	-	C-Strain	-	[111]
	Wild boars	Attenuated	-	C-Strain	-	[112]
AIV	Mice	Recombinant vector	100%	Baculovirus	H5N1 HA	[113]
	Chickens	Nanoparticle	-	Sliver nanoparticles	HA DNA	[114]
	Mice	Recombinant vector	100%	*Salmonella*	HA, NA	[115]
	Mice	Recombinant vector	100%	Yeast	H7N9 HA	[116]
	Chickens	Recombinant vector	100%	Yeast	H5N1 HA	[117]
	Chickens	Recombinant vector	100%	*Lactococcus lactis*	HA1 + M2	[118]
	Mice	Nanoparticle	100%	VLPs	H5N1 HA + VSP	[60]
PEDV	Mice, piglets	Recombinant vector	-	Yeast	S1	[119]
	Mice	Recombinant vector	-	*Lactobacillus casei*	COE	[63]
	Piglets	Recombinant vector	-	*Bacillus subtilis*	COE	[120]
	Mice	Recombinant vector	-	Adenovirus	COE + LTB	[121]
	Piglets	Plant-based	-	Maize grain	Spike protein	[83]
	Mice	Plant-based	-	*Nicotiana benthamiana*	COE-PIGs	[122]
	Mice	Recombinant vector	-	*Lactobacillus*	S1	[62]
*Clostridium tetani*	Mice	Nanoparticle	-	GM-bilosomes	Tetanus toxoid	[58]
*Campylobacter jejuni*	Chickens	Nanoparticle	-	PLGA	OMPs	[36]
Newcastle disease virus	Chickens	Attenuated	-	I2	-	[123]
*Clostridium perfringens*	Chickens	Plant-based	-	Tobacco	NetB, alpha-toxin, metallopeptidase	[124]
	Broilers	Recombinant vector	-	*Lactobacillus casei*	α-Toxin	[125]
Duck tembusu virus	Ducks	Recombinant vector	100%	*Salmonella*	PrM, E DNA	[126]
	Ducks	Recombinant vector	100%	*Salmonella*	Capsid protein DNA	[127]
Rabbit hemorrhagic disease virus	Rabbits	Recombinant vector	93.3%	*Salmonella*	VP60 DNA	[128]
Nipah virus	Pigs	Recombinant vector	-	Attenuated RABV	NiVG, NiVF	[129]
FMDV	Mice	Recombinant vector	-	*Lactococcus lactis*	VP1	[130]
	Guinea pigs	Nanoparticle	-	PLGA	VP1, VP3 DNA	[38]
*Brucella*	Mice	Recombinant vector	-	*Salmonella*	PrVgB	[131]
*Leptospira* spp.	Rats	Recombinant vector	-	*Salmonella*	LipL32	[132]
*Trichinella spiralis*	Mice	Recombinant vector	-	*Salmonella*	Ts43	[133]
*Mycoplasma hyopneumoniae*	Piglets	Nanoparticle	-	Silica SBA-15	Extraction proteins	[134]
IBDV	Hens	Plant-based	-	*Nicotiana benthamiana*	VP2	[135]
	Mice	Recombinant vector	-	Yeast	VP2	[136]
*E. coli*	Mice	Plant-based	-	Canola seeds	STxB, CfaB, LTB, Intimin	[82]
	Avian	Recombinant vector	100%	*Salmonella*	O-antigen	[137]
	Cattle	Attenuated	-	E16991,E16992,E16993	-	[138]
	Piglets	Modified subunit	-	αAPN-pIgA	FedF	[139]
*Salmonella*	Mice	Attenuated	100%	KST0556	-	[140]
	Rabbits	Attenuated	100%	HB1	-	[141]
	Chickens	Nanoparticle	-	Chitosan	OMPs, Flagellin protein	[142]
	Rhesus macaque	Attenuated	80%	CVD 1926	-	[143]
	Hens	Nanoparticle	-	PVM/MA	Heat extract fraction	[144]
	Chickens	Attenuated	80%	SG01	-	[145]
	Chickens	Attenuated	-	2S G10	-	[146]
	Broilers	Nanoparticle	-	Chitosan	OMPs, Flagella proteins	[53]
*Staphylococcus aureus*	Mice	Recombinant vector	-	*Salmonella*	rEsxAB, rHla_m_	[147]
*Mannheimia haemolytica*	Mice	Plant-based	-	*Nicotiana benthamiana*	LktA + PlpE	[148]
*Eimeria tenella*	Chickens	Recombinant vector	-	Yeast	EtAMA1, EtIMP1, EtMIC3	[149]
	Chickens	Recombinant vector	-	*Lactobacillus plantarum*	RON2	[150]
Goose parvovirus	Ducks	Nanoparticle	-	Cyclic peptide nanotubes	VP2 DNA	[151]
*Mycobacterium avium* subsp. *paratuberculosis*	Calves	Attenuated	-	*BacA*Δ	-	[152]

### 4.3. Oral Vaccines for Aquaculture

Recently, oral vaccines have been developed for a wide array of aquaculture fish species, predominantly targeting bacterial pathogens, with only a limited number designed to combat viruses. The method of vaccine administration varies depending on the age and size of the fish, with commercial vaccines administered orally (via incorporation into feed), through immersion (dip or bath), or via injection, either intraperitoneally (i.p.) or intramuscularly (i.m.) [153]. While injection vaccination typically yields the highest protection rates, it is accompanied by intensive handling and stress for the fish. Fish that are too small for injection are commonly vaccinated orally or through immersion, albeit these methods often result in lower efficacy and shorter-duration of protection. To ensure sustained protection throughout the entire production cycle, vaccination regimens have been devised for various species, employing a combination of immersion, oral, and injection vaccination strategies [154]. 

The fish oral vaccines that have been previously approved are primarily based on pathogens that have been killed through heat or formalin treatment. These vaccines target pathogens that are mostly intestinal, such as *Yersinia ruckeri* and *Vibrio anguillarum*, or those that enter and affect mucosal organs, such as *Piscirickettsia salmonis* and infectious pancreatic virus [154]. Commercial oral vaccines are available for Atlantic salmon and coho salmon against *P. salmonis*, IPNV, and infectious salmon anemia virus (ISAV), and for rainbow trout against IPNV, *Y. ruckeri*, and *P. salmonis*. Rainbow trout and sea bass have vaccines against *V. anguillarum* [153,155,156]. A single oral vaccine for Great amberjack against *Lactococcus garviae* has been licensed in Japan [154]. Despite their availability, oral vaccines for fish are predominantly used in a limited number of countries, mainly Chile, Norway, and Scotland [153]. 

Undoubtedly, oral administration stands out as the most practical method of immunization for aquatic vaccines, accommodating a broad spectrum of vaccine types, including inactivated, live attenuated, and nucleic acid vaccines. In contrast to injectable vaccination, oral administration offers flexibility independent of animal size and age, mitigates stress and physical trauma, yields labor cost savings, and aligns with animal welfare standards [157]. Given the intensive and factory-based nature of modern aquaculture, oral immunization emerges as the preferred choice for large-scale farm animals over alternative methods. Despite these advantages, the number of oral vaccines approved for use in the aquaculture sector remains sparse. For instance, among 17 commercially available vaccines targeting viruses in 2014, only 2 were oral formulations [155]. However, the widespread development and adoption of oral vaccines for fish have yet to materialize. Key constraints include vulnerability to enzymatic degradation in the digestive tract, inability to elicit both intestinal mucosal and systemic immune responses, and potential induction of immune tolerance. Moreover, challenges persist in ensuring the effective dosing and immunization of orally vaccinated fish, resulting in suboptimal immune protection compared to injection immunization. Nonetheless, ongoing research endeavors persist in the quest to develop oral vaccines capable of eliciting desired immune responses and conferring robust protection. The complexity of this pursuit is compounded by a dearth of comprehensive understanding regarding the parameters essential for eliciting effective immunological responses leading to protection, alongside the risk of inducing immune tolerance [158]. We have summarized the types and protection rates of oral vaccine candidates for aquaculture in Table 4.

### 4.4. Approved Oral Vaccine for Animals 

At present, only a restricted selection of oral vaccines for animals have secured market authorization, with specific examples outlined in Table 5. Predominantly, oral rabies vaccines dominate the landscape, showcasing significant efficacy in European efforts to combat rabies. Nonetheless, a notable case report has highlighted an instance wherein LYSVULPEN administration precipitated rabies development in a wild fox [172]. The scarcity of oral vaccines targeting various pathogens presently available on the market may stem from a confluence of factors, encompassing inadequate incentives for research and deployment, as well as a multitude of pertinent business considerations.

## 5. Conclusions and Future Perspectives

The utilization of oral vaccines for wildlife immunization against zoonotic diseases aligns with the One Health framework. Presently, there is limited research focusing on the field delivery of oral rabies vaccines for wildlife immunization. However, several generations of mature wildlife rabies oral vaccines, including RABORAL V-RG^®^, ONRAB^®^, and the SPBN GASGAS strains, have been successfully deployed in North America, Europe, and Africa, yielding significant outcomes in local rabies eradication efforts. Beyond wildlife oral immunization, stringent measures must be implemented to prevent and manage human exposure to oral vaccines, mitigating the risk of residual toxicity that could pose harm to both humans and domestic animals.

In addition to oral vaccines employed for rabies prevention, a substantial portion of oral vaccines for animals are geared towards thwarting enteric pathogens such as *E. coli*, *Salmonella*, and PEDV. In the case of these vaccines, the protective efficacy conferred by secretory IgA (sIgA) in the gut surpasses that of serum IgG, owing to the primary site of infection for enteric pathogens. Moreover, oral immunization offers a more cost-effective alternative to injectable vaccines. Currently, oral vaccines targeting enteric pathogens encompass oral inactivated, attenuated, vector, and subunit vaccines. Notably, the utilization of attenuated *Salmonella* to express exogenous antigenic proteins also elicits immunity against *Salmonella* itself, akin to the functionality of a bivalent vaccine but at a lower cost. Consequently, this approach represents an economically viable oral vaccine option for preventing intestinal infectious diseases in farm animals, possessing considerable research and application value both presently and in the future.

With advancements in vaccine technology, numerous innovations initially developed for injectable vaccines are now being progressively integrated into oral vaccine platforms. Of particular promise are oral subunit vaccines and oral recombinant replication-defective virus vector vaccines. These oral vaccines boast a high safety profile and minimal pathogenicity to both humans and animals, albeit necessitating robust adjuvants and delivery systems to confer adequate protection in animals. Significant research efforts have been directed towards investigating single-delivery carriers or adjuvants. An optimal oral vaccine delivery system should comprise multiple components, each fulfilling distinct functions such as sustained antigen release, resistance to gastric acid degradation, targeting of M cells, and augmentation of immune responses. The incorporation of novel antigens possessing potent immunogenicity holds the potential to elicit robust mucosal and systemic immune responses in animal hosts. Consequently, the development of delivery systems for oral vaccines and novel antigens assumes paramount importance in contemporary research endeavors and for future investigations.

Oral vaccines have lagged significantly behind parenteral vaccines in advancing. The most recently licensed oral vaccine was licensed more than a decade ago [173]. In addition, all currently available oral vaccines are based on earlier attenuated or inactivated vaccines. Outbreaks of COVID-19 in recent years and the continued emergence of new pathogens have created challenges and opportunities to update current vaccine knowledge and methods. Oral vaccination offers an option to augment or complement traditional parenteral routes. Its simplicity and relative cost-effectiveness can reduce logistical and economic burdens, while increasing the acceptance and compliance of the immunized individuals [173]. In addition, the combination of parenteral and mucosal immunity holds promise for improved protection. Therefore, efforts should be encouraged to further explore innovative strategies for oral or other mucosal vaccines.

## Figures and Tables

**Figure 1 vetsci-11-00353-f001:**
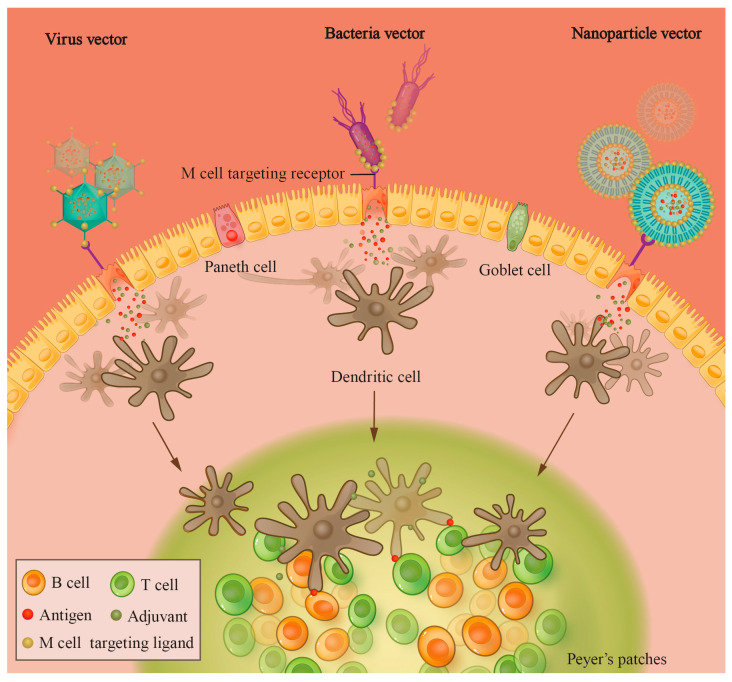
Schematic representation of the activation of the intestinal immune system by vectored oral vaccines. Viral/bacterial/nanoparticle vectors bearing targeting ligands for M cells can effectively localize and interact with M cells in the intestine, facilitating antigen endocytosis by M cells. After entering the intestinal lamina propria, antigens can be taken up by APCs such as DCs; DCs can migrate to the lymph node and induce the adaptive immune response.

**Table 1 vetsci-11-00353-t001:** Comparing the advantages and disadvantages of various oral vaccines.

Types of Oral Vaccines	Advantage	Disadvantage
Oral attenuated vaccine	Affordable and accessible, high immunogenicity	Virulence reversion risk, occasional side effects, low stability
Recombinant vectored oral vaccine	Good scalability, Multivalency [13]	Potential safety risks, pre-existing immunity (e.g., adenovirus vector) [13,14]
Nanoparticle oral vaccine	Control release, effect of adjuvants, biodegradable, can be designed rationally [15]	Complex production process, some potential risks of side effects and may transferred from biological barriers in animals [15]
Transgenic plant oral vaccine	Cost-effective and can be mass-produced in a green way [16], prolong the residence time on the mucosa [17], stable at room temperature [18]	Low expression level, risk of transgene contamination via pollen or seeds, need for oral priming with adjuvants [18]

**Table 4 vetsci-11-00353-t004:** Oral vaccine candidates for aquaculture.

Pathogen	Trial Animals	Type	Protection Rate	Strain/Vector	Antigen	References
*Aeromonas hydrophila*	Tilapia	Recombinant vector	70–100%	*Lactococcus lactis*	D1, D4	[159]
*Streptococcus agalactiae*	Nile tilapia	Attenuated	50–85%	YM001	-	[160]
	Tilapia	Recombinant vector	50–89%	*Lactococcus lactis*	SIP	[161]
*Edwardsiella ictaluri*	Channel Catfish	Attenuated	54–100%	S97-773	-	[162]
	Catfish	Attenuated	81–94%	NDKL1	-	[163]
Infectious pancreatic necrosis virus	Rainbow trout	Recombinant vector	-	*Lactobacillus casei*	CK6, VP2	[164]
	Rainbow trout	Recombinant vector	-	*Lactobacillus casei*	AHA1, CK6, VP2	[165]
Rock bream iridovirus	Rock bream	Recombinant vector	-	Yeast	Major capsid protein	[166]
	Rock bream	Plant-based	-	Rice callus	Major capsid protein	[167]
Nervous necrosis virus	Convict grouper	Nanoparticle	-	VLPs	-	[168]
Infectious pancreatic virus	Rainbow trout	Nanoparticle	80%	Alginates	VP2 DNA	[169]
Infectious hematopoietic necrosis virus	Rainbow trout	Nanoparticle	17–33%	PLGA	Glycoprotein DNA	[170]
	Rainbow trout	Nanoparticle	21–56%	Alginates	Glycoprotein DNA	[171]

**Table 5 vetsci-11-00353-t005:** Selected approved oral vaccines for animals.

Pathogen	Name of Medicine	Type	Company	Species
*Bordetella bronchiseptica*, Canine parainfluenza virus	Nobivac^®^ Intra-Trac^®^ Oral BbPi	Attenuated	Merck animal health	Canine
*Bordetella bronchiseptica*	TruCan B (Oral)	Attenuated	Elanco Animal Health	Canine
Rabies virus	Rabitec	Attenuated	CEVA Santé Animale	Foxes, raccoon dogs
	Rabigen SAG2	Attenuated	Virbac S.A.	Foxes, raccoon dogs
	Rabidog	Attenuated	Virbac S.A.	Canine
	RABORAL V-RG^®^	Recombinant vector	Boehringer Ingelheim	Raccoons, coyotes
	LYSVULPEN	Attenuated	Bioveta A.S	Foxes, raccoon dogs
*Salmonella typhimurium*	POULVAC^®^ ST	Attenuated	Zoetis	Poultry
Rotavirus	PROSYSTEM^®^ ROTA	Attenuated	Merck animal health	Swine
*Erysipelothrix rhusiopathiae*	ERY VAC 100	Attenuated	Arko Laboratories	Swine
*Lawsonia intracellularis*	Enterisol Ileitis	Attenuated	Boehringer Ingelheim	Swine
Rotavirus, coronavirus	Calf-Guard	Attenuated	Zoetis Animal Health	Calves
*Yersinia ruckeri*	AQUAVAC^®^ ERM Oral	Inactivated	Merck animal health	Rainbow trout

## Data Availability

No new data were created in this paper.

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
