# Peer review of "Recent Advances in Oral Vaccines for Animals"

_vetsci, 2024, doi:10.3390/vetsci11080353_

Round 1
Reviewer 1 Report
Comments and Suggestions for Authors
Compared to traditional injected vaccines, oral administrated vaccines offer significant advantages for the prevention of gastroenteric diseases. In the ms entitled “Recent Advances in Oral Vaccines for Animals”, Zhong et al reviewed the types of delivery strategies, influencing factors and applications of oral vaccines for animals, and highlights the global progress and achievement in oral vaccines, shedding light on potential future applications in this field. The topic of the manuscript is interesting. However, the ms was not well-organized and need major revision.
There are some weaknesses that need to be addressed.
1. It looks like the ms focus mainly on rabies vaccine. Vaccines for economic animals, such as pig, bovine, and horse are seldom mentioned. Or the title of ms can be minimized to focus on rabies oral vaccine development. In addition, some commonly used nanovaccine technologies, such as self-assembled nanovaccines, were not mentioned in the manuscript.
2. The ms is not well organized. A table summarizing the types, advantage, and disadvantage of oral vaccine is preferred in the introduction section for readers. The titles of section 2, 3, 4 cannot accurately summarize the contents of the section.
3. In section 2, for each of the delivery strategy for oral vaccine, several classical examples should be included at least for economic animals. A table to compare the difference between various delivery strategies would be helpful.
4. In section 3, factors influencing the efficacy of oral vaccines were full discussed, the content in the current version only provide some ways to improve efficacy. In addition to delivery strategies and vaccine type, the harsh environment of the gastrointestinal tract is the major challenge for oral vaccine, strategies addressing this issue should be discussed further.
5. Section 4 can be divided into three subsections, oral vaccines for aquaculture, for economic animals, and for urban stray animals and wildlife. In each subsection, different types of oral vaccines and classical example are discussed respectively.
Specialized note
1. Line 423, “4. ” is missed at the beginning of the topic.
Author Response
Comments 1: It looks like the ms focus mainly on rabies vaccine. Vaccines for economic animals, such as pig, bovine, and horse are seldom mentioned. Or the title of ms can be minimized to focus on rabies oral vaccine development. In addition, some commonly used nanovaccine technologies, such as self-assembled nanovaccines, were not mentioned in the manuscript.
Response 1: Thank you very much for raising this issue. We have included more information in the manuscript on the research progress and developments related to oral vaccines for livestock such as pigs, cattle, and poultry. These additions have been made mainly in Section 4.2 and Table 3. In addition, Table 2 now includes oral vaccines for wildlife other than rabies. With regard to nanovaccines, we have included a discussion of virus-like particles (VLPs)-based oral vaccines in Section 3.2.5 of the manuscript.
Comments 2: The ms is not well organized. A table summarizing the types, advantage, and disadvantage of oral vaccine is preferred in the introduction section for readers. The titles of section 2, 3, 4 cannot accurately summarize the contents of the section.
Response 2: We have added a table titled “Comparing the Advantages and Disadvantages of Various Oral Vaccines” (Table 1) to the “Introduction” section. Additionally, we have revised the titles of sections 2, 3, and 4 to better summarize their contents. The new titles are as follows: “2. Considerations for Designing Oral Vaccines for Animals”, “3. Delivery Strategies and Vectors of Oral Vaccines” and “4. Utilizations of Oral Vaccines in Various Animals”.
Comments 3: In section 2, for each of the delivery strategy for oral vaccine, several classical examples should be included at least for economic animals. A table to compare the difference between various delivery strategies would be helpful.
Response 3: In the revised manuscript, we have moved the original Section 2 to become Section 3. In the new Section 3, we have added a subsection titled “3.1. Classical Delivery Strategies,” which discusses the immunization of animals using traditional oral vaccines (live attenuated vaccines) administered through feeding and drinking water. Since different delivery strategies for vaccines essentially represent different types of vaccines, we have integrated the comparison of their advantages and disadvantages into the new Table 1.
Comments 4: In section 3, factors influencing the efficacy of oral vaccines were full discussed, the content in the current version only provide some ways to improve efficacy. In addition to delivery strategies and vaccine type, the harsh environment of the gastrointestinal tract is the major challenge for oral vaccine, strategies addressing this issue should be discussed further.
Response 4: We have moved the original Section 3 to the front, renaming it Section 2. The new title for Section 2 is “Considerations for Designing Oral Vaccines for Animals.” Additionally, we have added a new subsection, “2.1. Challenges of the Harsh Gastrointestinal Environment,” to address the issue in more detail.
Comments 5: Section 4 can be divided into three subsections, oral vaccines for aquaculture, for economic animals, and for urban stray animals and wildlife. In each subsection, different types of oral vaccines and classical example are discussed respectively.
Response 5: Thank you very much for your constructive feedback. We have revised Section 4 as follows: “4.1. Oral Vaccines for Urban Stray Animals and Wildlife”, “4.2. Oral Vaccines for Economic Animals”, “4.3. Oral Vaccines for Aquaculture”, and “4.4. Approved Oral Vaccines for Animals”. We have also updated the corresponding tables accordingly.
Comments 6: Line 423, “4. ” is missed at the beginning of the topic.
Response 6: Thank you for pointing out this oversight. We apologize for the mistake. The “4.” has now been added at the beginning of the corresponding heading.
Reviewer 2 Report
Comments and Suggestions for Authors
This is a good review which gives a good overview over oral vaccination. But a few things NEED to be improved. As they are easily done, I recommend minor revisions but expect these additions to be done.
1) The authors should explain the mode of action of classical oral vaccines, in particular rabies. Does the attenuated vaccine really induce intestinal immunity or is immunity caused by e.g. local oral infection. This is a very critical issue for designing future oral vaccines in general.
2) Oral vaccination also induces IgG
3) IgA does not mediate ADCC effectively. IgG does.
4) I do not understand Fig 1. Needs more explanation.
5) Section 2.1 How does any of this stabilize the antigen that is to be immunized against? It seems these measures stabilize the delivery vehicle rather than the antigen.
Author Response
Comments 1: The authors should explain the mode of action of classical oral vaccines, in particular rabies. Does the attenuated vaccine really induce intestinal immunity or is immunity caused by e.g. local oral infection. This is a very critical issue for designing future oral vaccines in general.
Response 1: Thank you very much for your constructive suggestion. According to the literature, the immunogenic effect of the oral attenuated rabies vaccine is significantly associated with the vaccine virus’s ability to infect the tonsils. Therefore, the tonsils represent a critical target organ to consider in oral vaccine delivery strategies. We have added relevant content and references in the second paragraph of the new section “3.1. Classical Delivery Strategies”.
Comments 2: Oral vaccination also induces IgG
Response 2: Thank you for pointing this out. We have revised the original sentence to: “The antibodies produced by mucosal humoral immunity following oral vaccination are mainly secretory IgA (sIgA) and IgG.”
Comments 3: IgA does not mediate ADCC effectively. IgG does.
Response 3: Based on the literature, including Zahavi, David et al. “Enhancing antibody-dependent cell-mediated cytotoxicity: a strategy for improving antibody-based immunotherapy.” Antib Ther. 2018;1(1):7-12, and Davis, Samantha K et al. “Serum IgA Fc effector functions in infectious disease and cancer.” Immunol Cell Biol. 2020;98(4):276-286, both IgG and IgA can trigger ADCC by binding specifically to FcγR and FcαR, respectively. However, the ADCC effect of IgA is primarily associated with anti-tumor responses. To avoid any potential confusion, we have decided to remove the related content. The revised sentence is: “sIgA is essential for the development of the mucosal barrier, being predominantly located in the outer layer of the mucosa. Its primary functions include binding and clustering of pathogens, blocking their entry into mucosal epithelial cells, restricting their movement, and eliminating pathogens present in the mucosa.”
Comments 4: I do not understand Fig 1. Needs more explanation.
Response 4: Figure 1 primarily illustrates the activation of the intestinal immune system by vectored oral vaccines. To clarify the purpose of Figure 1, we have revised the title and added a more detailed caption.
Comments 5: Section 2.1 How does any of this stabilize the antigen that is to be immunized against? It seems these measures stabilize the delivery vehicle rather than the antigen.
Response 5: Overall, these nanoparticles can encapsulate antigens, protecting them from the acidic environment and proteases in the gastrointestinal tract. Additionally, these materials are highly amenable to modification or incorporation of adjuvant components, which enhances their targeting capability and improves the efficacy of oral immunization. To address this concern, we have added relevant content at the beginning of Section 3.2 (formerly Section 2.1).
Round 2
Reviewer 1 Report
Comments and Suggestions for Authors
My concerns have been addressed.